# Advancing Wildlife Conservation Through Biobanking in South America

**DOI:** 10.3390/ani15223261

**Published:** 2025-11-11

**Authors:** Carla B. Madelaire, Alexsandra F. Pereira, Adrián J. Sestelo, Aléxia P. Bom-Conselho, Carolina Vaj, Felipe C. Mosalve, Larissa S. Brandão-Souza, Marcela R. Tavares, Matteo Duque Rodriguez, Raquel O. Restrepo, Roberta F. Leite, Yann Locatelli, Thyara Deco-Souza, Gediendson R. de Araujo

**Affiliations:** 1Beckman Center for Conservation Research, San Diego Zoo Wildlife Alliance, Escondido, CA 92027, USA; cbonettimadelaire@sdzwa.org; 2Laboratory of Animal Biotechnology, Federal Rural University of Semi-Arid, Mossoró 59625-000, RN, Brazil; alexsandra.pereira@ufersa.edu.br; 3Laboratorio de Biotecnología Reproductiva para la Conservación Animal y Biobanco, Ecoparque Interactivo, Buenos Aires C1425, Argentina; adriansestelo@gmail.com; 4Instituto Reprocon, Campo Grande 79052-280, MS, Brazil; alexiapbc@gmail.com (A.P.B.-C.); larissasbs@usp.br (L.S.B.-S.); bobbieleite@gmail.com (R.F.L.); thyara.araujo@ufms.br (T.D.-S.); 5Faculty of Veterinary Medicine and Animal Science, University of São Paulo, São Paulo 05508-040, SP, Brazil; 6Grupo INCA-CES, Facultad de Medicina Veterinaria y Zootecnia, Universidad CES, Medellín 050001, Antioquia, Colombia; caritovaj26@gmail.com (C.V.); fmonsalve@ces.edu.co (F.C.M.); mduquero@ces.edu.co (M.D.R.); raquelochoa8r@gmail.com (R.O.R.); 7Grupo de Investigación en Biotecnología Animal, Facultad de Ciencias Agrarias, Politécnico Colombiano Jaime Isaza Cadavid, Medellín 050021, Antioquia, Colombia; 8BioParque do Rio, Quinta da Boa Vista, São Cristóvão, Rio de Janeiro 20940-040, RJ, Brazil; marcela.tavares@grupocataratas.com; 9Réserve Zoologique de la Haute Touche, Muséum National d'Histoire Naturelle, 36290 Obterre, France; yann.locatelli@mnhn.fr; 10Postgraduate Program in Animal Science, Faculty of Zootechnics and Veterinary Medicine, Federal University of Mato Grosso do Sul, Campo Grande 79070-900, MS, Brazil

**Keywords:** wildlife conservation, ex situ conservation, assisted reproductive technologies, conservation policies

## Abstract

**Simple Summary:**

Biobanks are special collections that keep animal cells, tissues, and genetic material stored under safe conditions for long periods of time. In South America, this is very important because the region has one of the greatest biodiversities in the world, but several wild animals are facing serious threats such as habitat loss, climate change, and illegal hunting. By saving biological samples today, biobanks create opportunities to study, protect, and even restore species in the future. This review explores how biobanking is developing in South America, the progress already achieved, and the difficulties that still need to be overcome. It also highlights how connecting biotechnology with conservation can open new paths to protect endangered wildlife and ensure that the region’s natural heritage is preserved for the generations.

**Abstract:**

South America harbors one of the world’s richest biodiversities, yet its wildlife faces escalating threats from climate change and anthropogenic pressures. Biobanking different types of cells and tissues represents an important strategy to preserve genetic diversity and support conservation efforts in the long run. This review highlights the main challenges, opportunities, and future perspectives for biobanking as a conservation tool in South America. Key challenges include technical standardization, funding, and integration with conservation policies. Despite these barriers, recent advances demonstrate the growing importance of biobanking as a complementary tool for safeguarding endangered species and strengthening long-term conservation strategies in the region. The integration of biotechnological approaches into conservation programs positions biobanks as pivotal tools for advancing wildlife management and safeguarding the unique biodiversity of South America.

## 1. Introduction

In the past century, vertebrate extinction rates have surpassed natural levels, marking the first human-driven mass extinction—the sixth in 65 million years. According to the International Union for Conservation of Nature (IUCN), 41% of amphibians, 26% of mammals, and 14% of birds face extinction [1,2]. Urgent conservation strategies, both in situ and ex situ, are critical to mitigate biodiversity loss and preserve genetic diversity [3,4].

In recent years, biobanking has been shown to be a robust complementary conservation strategy to preserve genetic diversity that is essential for species resilience (Figure 1). Its value is now widely recognized as a core component of the One Plan and One Health approaches (https://www.cpsg.org/our-work/our-approach/one-plan-approach, accessed on 24 October 2025) and multiple international frameworks, including the United Nations Sustainable Development, the Convention on Biological Diversity, the Nagoya Protocol, and the Kunming-Montreal Global Biodiversity Framework, with the IUCN formally forming a Conservation Specialist Group in the topic (https://iucn.org/our-union/commissions/group/iucn-ssc-animal-biobanking-conservation-specialist-group, accessed on 24 October 2025). Animal biobanks can indefinitely store several sample types, including live somatic cell lines, gametes, embryos, and non-live samples, such as tissues, DNA, exosomes under deep cold (−196 °C) (see Table 1, Figure 1) [5,6,7,8].

When accessible to researchers, these repositories can be powerful tools for effective conservation biotechnology and other applications that can help manage and recover in situ and ex situ populations [16] (Figure 1). Thus, biobanks are active platforms for conservation, beyond their traditional role as passive repositories, integrating sample storage with advanced reproductive technologies, genomic analysis, and adaptive management strategies [17] (Figure 1).

However, building a biobank encompasses several challenges, including collecting meaningful biological samples, developing standardized protocols, and building and maintaining worldwide networks [7,18,19,20] (Figure 1). South America, the home of several biodiversity hotspots, offers vast opportunities for enhancing biobank conservation efforts in the region [21]. This document discusses the challenges, opportunities, and future directions of biobanking for wildlife conservation in South America.

## 2. Challenges in Establishing Biobanks for Conservation in South America

A core challenge in biobanking lies in the lack of standardized, clear, and repeatable protocols for sample collection, preservation, and validation. There is currently no established consensus on the priority for creating biobanks for a species over others or for specific individuals within a population. However, the threat level of the species, as presented by the IUCN Red List [2], could be the basis for establishing these priorities. Other factors, such as lower genetic diversity of the population, and a decreasing trend in the number of individuals in the population, should also be considered. It is important to collect material from populations that are still classified as Near-Threatened or Vulnerable, as they have greater genetic amplitude [22,23]. The relevance of the species to the ecosystem and knowledge related to the reproduction and reproductive biotechnologies of the species are also highly relevant [22].

The development of standardized protocols that align with international best practices while also being adaptable to local contexts guarantees that the material stored remains viable in the long term. For somatic cells, factors like tissue origin [24], morphological differences [25], and transport conditions [26] might demand tailored approaches. Germplasm biobanking is even more complex, particularly for female gametes and non-mammalian taxa (e.g., amphibians, reptiles, sharks), where the absence of domestic models and several logistical barriers—such as the remoteness of wild populations, limited access to specialized equipment, and the difficulty of maintaining appropriate temperature and transport in field conditions—complicate protocol development [22,27]. For instance, reptiles show a temperature-dependent gamete viability that requires customized protocols [28]. For amphibians, the world’s most threatened vertebrates, reproductive biotechnology protocols have been developed, including semen cryopreservation in *Atelopus spumarius* [29] and a simplified intracytoplasmic sperm injection (ICSI) protocol in *Xenopus* [30]. Several groups in South America have made progress in unveiling variability across cell types, species, and individuals, adapting equipment and protocols traditionally used for domestic animals [27,31]. However, much more work has to be done to keep advancing efforts in the region.

Many endangered species lack ex situ insurance populations, making field collection essential, adding another layer to the challenges of biobanking genetic diversity. For example, Brazil’s critically endangered jaguar populations (from Caatinga and Atlantic Forest) have no ex situ individuals, resulting in the need to collect tissue in free-living and *post-mortem* sampling strategies [32,33,34]. Reproductive and skin tissues obtained from *post-mortem* animals in advanced decomposition may be challenging to establish cell culture. In cases where the animal has been dead for many hours, and the tissue is in an advanced state of decomposition, it can be functionally restored by xenotransplantation into immunodeficient mice [35], offering a valuable conservation tool for difficult-to-capture species frequently found as roadkill. The challenges associated with capturing animals in the wild and keeping samples viable until they can be processed in the lab require multiple collaborations between different teams and organizations. While immediate refrigeration and/or freezing might seem ideal, maintaining proper cryogenic conditions in remote field sites is often unfeasible, and temperature fluctuations during transport can compromise cell viability and tissue integrity. Furthermore, many tissues and gametes require short-term processing or equilibration to ensure successful long-term preservation.

Additionally, the decision-making process for species prioritization that considers IUCN threat status with genetic diversity and ecological role can be hampered because of the lack of data for several species from understudied areas [22,23]. Mammals are the majority of taxa represented in biobanking efforts, while reptiles, amphibians, and fish remain underrepresented [23]. This gap reflects both technical hurdles and limited funding for non-charismatic species. A regional strategy could address this by targeting evolutionarily distinct or climate-vulnerable taxa, leveraging South America’s megadiverse ecosystems. 

Biobanks are only as valuable as their curation and associated data, yet many South American collections lack standardized databases. For storage, a secure, controlled environment with adequate space is required, along with redundancy through duplicated locations and backup systems, continuous temperature maintenance (see Table 1 for requirements for each sample type), regular equipment maintenance, sample integrity monitoring, and a robust labeling system to ensure proper tracking, while allowing for future expansion. Effective management also requires meticulous sample metadata (e.g., collection protocols, animal health records), and integration with other datasets (e.g., physiology, genomics, proteomics). Consistent sample annotations using international standards are vital for these samples to be usable for conservation purposes [36,37,38,39]. Lessons from human biobanks could guide infrastructure development, but conservation-specific adaptations, such as ecological and behavioral data linkages, could be potentially vital for future conservation efforts [40,41,42]. A significant challenge to data sharing is the absence of a universal, accessible database for all countries and users. In the United States, initiatives like Zoological Information Management Systems (ZIMS; species360.org) have started developing standardized data modules to capture metadata. However, annual membership fees can be prohibitive for many South American institutions. Consequently, an open-access database is vital to ensure equitable accessibility. Additionally, the lack of clear legislation addressing genetic resource ownership, sample exportation, and health monitoring in South America is also a challenge to advancing biobanking. Unlike human biobanking, which is well-regulated in many countries (e.g., the EU’s General Data Protection Regulation [GDPR] and the U.S. NIH guidelines), wildlife biobanking remains largely unregulated in the region. Brazil, despite its unparalleled biodiversity, still lacks specific laws governing wildlife biobanks, relying instead on fragmented regulations from agencies like Brazilian Agricultural Research Corporation (EMBRAPA, which oversees agricultural genetic resources) and Chico Mendes Institute for Biodiversity Conservation (ICMBio, responsible for endangered species). Some countries, such as the UK (Frozen Ark Project) and Australia (Australian Wildlife Biobank), have advanced policies that could serve as models. However, in South America, only a few initiatives, such as Argentina’s Banco de Germoplasma de Fauna Silvestre have begun formalizing biobanking practices. Given the region’s role as a biodiversity hotspot, governments must urgently develop tailored legislation to regulate ethical standards while facilitating conservation research. Without legal clarity, biobanking efforts risk inefficiency, ethical conflicts, and missed opportunities to safeguard genetic diversity in the face of escalating extinction threats.

Finally, biobanking’s potential hinges on coordinated networks. South America’s biobanks initiatives are often fragmented across universities, zoos, and Non-Governmental Organizations (NGOs), with uneven adherence to standards. Building a resilient network requires harmonizing protocols and advertising guidelines, securing long-term funding, and fostering cross-border partnerships, especially for species spanning multiple countries (e.g., Andean condors). The new IUCN’s Animal Biobanking Specialist Group (2022) (https://www.iucn.org/our-union/commissions/group/iucn-ssc-animal-biobanking-conservation-specialist-group, accessed on 18 March 2024) offers a starting point, but regional buy-in and developing local policy to adapt to the South American scenario are essential to ensure biobanks become a cornerstone of conservation in the region. This regional approach would not only strengthen the individual conservation efforts of each country but also provide a robust, interconnected network that is more resilient to future challenges in wildlife conservation.

## 3. Opportunities

Biobanks can significantly boost biodiversity conservation through strategic collaborations. Partnerships with different institutions and laboratories accelerate research by sharing knowledge, resources, and infrastructure [43]. Regional networks are especially valuable, enabling countries to share data, methods, and expertise while leveraging niche knowledge in species and ecosystems. Such cooperation enhances biobanks’ efficiency, scalability, and impact. Training and capacity-building of professionals involved in biobank operations and management is another missed opportunity that can help standardize techniques across institutions, ensuring the viability of stored samples. Additionally, teamwork and the inclusion of diverse professional profiles are crucial for addressing challenges in an interdisciplinary manner. Investing in training not only improves biobank efficiency but also strengthens the network of professionals dedicated to conservation, creating a solid foundation for innovative solutions.

Technological innovations such as cloning, cellular reprogramming and genomics in phenome research open new possibilities in the field of biobanking [9,12,44]. These techniques have transformative potential, both for the conservation of endangered species and for regenerative medicine, allowing, for example, the recovery of genetically viable populations from stored samples [4,45,46].

The lack of financial resources is a significant obstacle, but also an opportunity to engage new stakeholders by demonstrating the value of biobanks for biodiversity conservation [47,48]. There is an opportunity to grow, maintain, and consolidate biobank practices in the region. Developing robust plans and securing diversified sources of funding, including public and private, as well as partnerships with non-governmental organizations, is probably the most well-structured strategy.

Ensuring responsible and ethical use of biological data requires prioritizing both data sharing and transparency [49]. While raw data may contain sensitive information, aggregated datasets are often sufficient to inform decision-making for practitioners, policymakers, and the public. To achieve this balance, the establishment of ethical data-sharing frameworks is critical, particularly to build trust with Indigenous communities and other stakeholders who provide sensitive or proprietary information (Nagoya protocol—http://data.europa.eu/eli/agree_prot/2014/283/oj, accessed on 24 October 2025). Promoting structured, transparent, and collaborative data practices allows biobanks to maximize their scientific and conservation impact while safeguarding privacy and equity. Effective database planning and management should foster a symbiosis between the biobank operations and the IT systems to ensure maintenance of samples, facilitate data collection, and support conservation efforts and future research and international cooperation [37,38]. Finally, incorporating biobanking as a topic in higher education and raising public awareness by engaging local communities, research centers, and universities creates a shared vision of the importance of this conservation practice that can foster a long-term culture of protection and responsibility toward biological heritage [50].

## 4. The Future of Biobanking in South America

In recent years, the expansion of biobanks has completely transformed the way biodiversity is conserved [51]. In the short to medium term, biobanks preserve and help identify genetic diversity, which supports targeted breeding and other conservation strategies. This is especially vital in biodiverse regions like Latin America, where the rate of habitat loss and climate change is significantly higher [52].

A remarkable example of successful conservation is Brazil’s program for the golden lion tamarin (*Leontopithecus rosalia*), a critically endangered primate endemic to the Atlantic Forest. In 1960, the population was just 300 individuals. Then, coordinated efforts in semen cryopreservation, genetic management, and assisted reproduction boosted numbers to 4800 by 2023 [53] (Arakaki et al., 2019 [54]; https://www.savetheliontamarin.org/news/population-census-shows-31-increase-in-wild-glts, accessed on 24 October 2025). By integrating biobanking with captive breeding and reintroduction programs, supported by partnerships among several institutions, researchers have not only recovered population numbers but also preserved genetic diversity. This approach demonstrates how biobanks can provide ethical and sustainable models to restore threatened populations while minimizing wild captures [55]. A pioneering study in Argentina with bobcats (*Lynx rufus*) demonstrated the potential of reproductive biotechnologies for wild feline conservation. Researchers successfully cryopreserved semen and evaluated their fertility using domestic cat oocytes in a heterologous in vitro fertilization, producing hybrid embryos that developed to the pre-implantation stage [56]. While these findings represent an important step toward the use of assisted reproductive technologies (ARTs) in wild felids, the results remain preliminary. Significant challenges persist, particularly in obtaining viable and mature bobcat oocytes consistently and ensuring the production of biologically competent embryos when using conspecific gametes. Therefore, further research is required before these techniques can be reliably applied to support sustainable population management in this species.

New technologies need to be explored to overcome many of the logistical limitations of traditional cryopreservation. One promising example is dry biobanking, an alternative inspired by natural mechanisms such as anhydrobiosis, which involves the preservation of cells through controlled desiccation. This technique could enable the storage of genetic material at room temperature, with lower costs, reduced dependence on specialized infrastructure, and greater safety during transport. Its potential is particularly valuable for countries with technical or economic constraints, as it democratizes access to genetic conservation technologies and expands their global reach [57]. However, the practical application of dry biobanking still faces important limitations. To produce viable offspring from desiccated material, advanced assisted reproductive technologies (ARTs) such as intracytoplasmic sperm injection (ICSI) and embryo transfer must be optimized for each target species. The infrastructure, technical expertise, and species-specific biological knowledge required for these procedures remain limited in many regions, highlighting the need for further research and capacity building before large-scale implementation.

The integration of biobanking with assisted reproductive technologies presents a powerful conservation solution for endangered species. By combining cryopreservation, genomic-phenotypic analysis, and assisted reproduction, we can restore populations even from critically diminished genetic pools. These technologies can provide crucial insights into evolutionary adaptations and genetic diversity, essential tools for combating habitat loss and climate change impacts (Howell et al., 2022 a,b [58,59]). Modern biobanks should be more than genetic repositories and act as dynamic conservation platforms, preserving biodiversity while helping enhance ecological resilience, which is a vital tool in addressing the global biodiversity crisis ([4,60,61], Howell et al., 2020 [62]; https://reviverestore.org/projects/biobanking/, accessed on 24 October 2025).

Looking ahead, biobanks are evolving into standardized, collaborative systems with shared protocols and ethical frameworks. Public–private and non-profit partnerships could help drive innovation while ensuring financial and operational sustainability. To overcome potential ethical risks when profit motives influence the use of biomaterials from endangered species, implementing transparent governance and strict material transfer agreements is essential to prevent misuse and ensure that such collaborations align with conservation goals. Additionally, the convergence of assisted reproductive technologies, genome-to-phenome research, and biobanking offers a promising outlook for the conservation of endangered species, making it possible to restore viable populations from stored genetic material, even in scenarios of low genetic variability or severe population decline.

Finally, the future of biobanks is projected toward a more integrated, harmonized, and sustainable model. This entails progress toward the standardization of protocols, data interoperability, and the implementation of common regulations to ensure the quality, traceability, and ethical use of biological resources and improve technological innovation, capacity building, and financial sustainability.

## 5. Conclusions

Biobanking represents a powerful tool to strengthen wildlife conservation strategies in South America, a region recognized for its biodiversity but also for its high level of threats to its ecosystems. By enabling the preservation of genetic, cellular, and reproductive resources, biobanks provide opportunities for research, species recovery, and long-term safeguarding of genetic diversity. However, advancing this field requires coordinated efforts among governments, academic institutions, conservation organizations, and local communities. Establishing standardized protocols, securing sustainable funding, and promoting international collaboration will be essential to fully unlock the potential of biobanking in South America.

## Figures and Tables

**Figure 1 animals-15-03261-f001:**
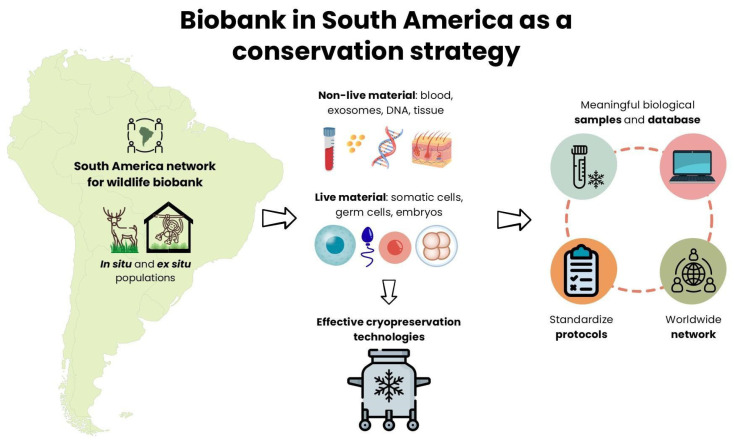
Biobank as a conservation strategy consists of the integration of diverse and viable sample types accompanied by relevant metadata, along with in situ and ex situ conservation strategies coordinated at both regional and world-wide levels.

**Table 1 animals-15-03261-t001:** Types of samples and their definition, preservation temperature and examples of application.

Sample Type	Definition	Live or Non-Living	Preservation Temperature	Possible Application	Examples of Conservation Application	Reference
Somatic cell	Diploid cells (e.g., fibroblasts)	Live	Liquid nitrogen (−196 °C)	Experimental cell physiology	Jaguar (*Panthera onca*) fibroblasts induced to pluripotent stage	[9]
Germ cells	Reproductive cells (e.g., sperm, oocytes)	Live	Liquid nitrogen (−196 °C)	Assisted Reproduction	Cryopreservation of maned wolf (*Chrysocyon brachyurus*) sperm	[10]
Tissues, blood, and cell pellet	Aliquots of organs, blood, non-live cells	Non-living	Freezer (−80 °C)	Genomic sequencing	Several species	[11]
Embryos	Early-stage development of multicellular organisms produced by IVF or cloning	Live	Liquid nitrogen (−196 °C)	Assisted Reproduction	Neotropical deer species	[12]
Exosomes	Messager particles released by cells	Non-living	Freezer (−80 °C)	Therapeutic application	Several species	[13]
Microbiomes	Associated micro-organisms (bacteria, fungi, viruses)	Live	Liquid nitrogen (−196 °C)/Freezer (−80 °C)	Conservation and restoration of the adaptive capacity of a species to its environment	General	[14,15]

## Data Availability

The data presented in this study are available upon request from the corresponding author.

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
