# Peer review of "Advancing Wildlife Conservation Through Biobanking in South America"

_animals, 2025, doi:10.3390/ani15223261_

Round 1
Reviewer 1 Report
Comments and Suggestions for Authors
This review summarizes the status of biobanking in South America and discusses the challenges and opportunities that exist currently. This manuscript is well written for the most part, but there are sections that could benefit from English language correction. Additionally, the authors seem to oversimplify the challenges facing the development of biobanks for endangered species. Authors are requested to kindly review the comments below and make the necessary revisions.
Specific comments:
Line 44-46: This sentence warrants a rewrite to provide clarity.
Table 1: Under Exosomes, change ‘cels’ to ‘cells’.
Table 1: Under ‘Microbiomes’, change ‘viroses’ to ‘viruses’.
Line 92: Suggest changing ‘change how we do conservation in the region’ to ‘enhance conservation efforts in the region’.
Line 92-94: This sentence warrants a rewrite.
Under the sub-section ‘Challenges in Establishing Biobanks for Conservation in South Amerca’ the authors have overlooked an important topic. Recently, researchers and zoological organizations from around the world have initiated efforts to establish biobanks. However, only a handful have clearly articulated the objectives and scope of these biobanks. As it is now evident, not everyone can establish and maintain functional biobanks. Simply collecting and storing samples is not biobanking. Programs must define what are the purposes of these biobanks? What taxa or species will be collected and banked should also be defined. What about the infrastructure investments required for establishing a biobank? These are some of the important issues that anyone interested in establishing a biobank should consider. Also, what is the long-term plan for these biobanks? There are several examples where a biobank has been set up by a single individual and then when that individual leaves the institution, these collections become orphaned. These are also some of the major challenges that should be discussed before considering technical details of collection and storage of biological samples.
Line 104: Authors are requested to expand on ‘logistical barriers’.
Line 113: Authors are requested to revise this sentence ‘Many endangered species lack human-cared populations, making field collection essential…’. For example, Many endangered species lack ex situ insurance populations…’.
Lines 114-117: Is this sentence correct? This reviewer is aware of several Zoos and Conservation Centers in Brazil that manage jaguars in human care. This sentence warrants some clarification. For example, authors are requested to check census information in zoological databases such as ZIMS.
Lines 117-120: Why should skin biopsies be xeno-transplanted to generate fibroblast cell lines?
Lines 120-123: This sentence should be further expanded. For example, why can’t tissue biopsies be maintained cold under field conditions until processing in the lab? Authors state that ‘require a partnership between different teams and organization…’. What do the authors mean by this? Authors are requested to provide additional clarifications in the manuscript.
Lines 126-139: As stated earlier, these sentences could be moved to the top of this section to highlight the importance of defining priorities for the biobank and for articulating the long-term objectives of a biobank.
Lines 141-143: The authors state ‘Effective management requires appropriate storage, meticulous sample metadata…’. It would be good to first highlight the importance of metadata and then storage conditions. Further, the authors are also requested to expand on ‘storage’ requirements.
Lines 202-204: A major challenge to data sharing is the lack of a universal database that is available to all countries and users. Even in advanced countries such as the United States, not all institutions involved in biobanking use the same database. Recently, zoological information systems (ZIMS; species360.org) has initiated the development of data modules to capture metadata in a standardize manner. However, these resources require annual membership fees that most institutions may not be able to support. Therefore, standardization of metadata and development of an open access (free) database is also warranted.
Lines 209-212: Authors are requested to rewrite this sentence to provide improved clarity.
Lines 224-231: These statements are not supported by appropriate citations. This reviewer is not aware of any reports where semen from golden lion tamarins have been systematically collected, cryopreserved, and banked. Further, there are no reports of successful development and application of ARTs in this species for species recovery. To this reviewer’s knowledge, the current success was achieved primarily via stringent genetic management and natural breeding. Authors are requested to provide additional evidence to support these statements.
Lines 233-240: The authors seem to over interpret some of the preliminary results that have been published in support of ARTs in bobcats. The mere production of domestic cat x bobcat hybrid embryos does not guarantee the consistent production of biologically competent bobcat embryos when frozen bobcat sperm is used to fertilized bobcat oocytes. Challenges remain with respect to obtaining viable, mature bobcat oocytes consistently. Authors are requested to acknowledge these remaining challenges before claiming that these results could address current challenges in maintaining sustainable populations.
Lines 241-248: Authors are requested to discuss potential limitations of dry biobanking as well here. For example, with dry biobanking, more advanced ARTs including ICSI and embryo transfer also must be developed for target species for production of live offspring. This information (and access to ICSI infrastructure) is still lacking for a large number of species and in several countries.
Lines 259-260: The concept of ‘public-private’ partnerships in development of biobanks is often discussed. But it is imperative to discuss potential risks associated with such partnerships. Private entities are often driven by revenue generation or profits. But the use of biomaterials from endangered species for revenue generation raises significant ethical concerns. Emphasis should also be placed on development of strict material transfer agreements to limit the use of these biological materials. Authors are requested to discuss these concerns and suggest solutions to mitigate these concerns in this section.
Author Response
This review summarizes the status of biobanking in South America and discusses the challenges and opportunities that exist currently. This manuscript is well written for the most part, but there are sections that could benefit from English language correction. Additionally, the authors seem to oversimplify the challenges facing the development of biobanks for endangered species. Authors are requested to kindly review the comments below and make the necessary revisions.
Specific comments:
Line 44-46: This sentence warrants a rewrite to provide clarity.
Answer: The sentence was rewritten.
Table 1: Under Exosomes, change ‘cels’ to ‘cells’.
Answer: Done.
Table 1: Under ‘Microbiomes’, change ‘viroses’ to ‘viruses’.
Answer: Done.
Line 92: Suggest changing ‘change how we do conservation in the region’ to ‘enhance conservation efforts in the region’.
Answer: Thank you for the suggestion. The sentence has been altered.
Line 92-94: This sentence warrants a rewrite.
Answer: The sentence was rewritten.
Under the sub-section ‘Challenges in Establishing Biobanks for Conservation in South Amerca’ the authors have overlooked an important topic. Recently, researchers and zoological organizations from around the world have initiated efforts to establish biobanks. However, only a handful have clearly articulated the objectives and scope of these biobanks. As it is now evident, not everyone can establish and maintain functional biobanks. Simply collecting and storing samples is not biobanking. Programs must define what are the purposes of these biobanks? What taxa or species will be collected and banked should also be defined. What about the infrastructure investments required for establishing a biobank? These are some of the important issues that anyone interested in establishing a biobank should consider. Also, what is the long-term plan for these biobanks? There are several examples where a biobank has been set up by a single individual and then when that individual leaves the institution, these collections become orphaned. These are also some of the major challenges that should be discussed before considering technical details of collection and storage of biological samples.
Answer: Thank you for your thoughtful and constructive comments. We appreciate your emphasis on the need to clearly define the objectives, scope, and long-term sustainability of conservation biobanks. In the revised version, we have incorporated a discussion on these critical aspects, including the importance of setting clear goals, identifying target taxa, assessing infrastructure requirements, and ensuring institutional continuity to prevent the loss of collections. Our aim in this review is to make biobanking standards more accessible and widely understood, helping to clarify these complexities and promote responsible, well-planned implementation of biobanking initiatives. We appreciate your valuable input, which has helped us strengthen this manuscript.
Line 104: Authors are requested to expand on ‘logistical barriers’.
Answer: The sentence was rewritten.
Line 113: Authors are requested to revise this sentence ‘Many endangered species lack human-cared populations, making field collection essential…’. For example, Many endangered species lack ex situ insurance populations…’.
Answer: Thank you for the suggestion. The sentence has been altered.
Lines 114-117: Is this sentence correct? This reviewer is aware of several Zoos and Conservation Centers in Brazil that manage jaguars in human care. This sentence warrants some clarification. For example, authors are requested to check census information in zoological databases such as ZIMS.
Answer: What we are asserting with this statement is that there are no individuals that originated from the Caatinga and Atlantic Forest biomes. Since the populations of each biome are phenotypically different (consisting of different ecotypes), the ex situ population must represent each of these genetic varieties.
Lines 117-120: Why should skin biopsies be xeno-transplanted to generate fibroblast cell lines?
Answer: Xenotransplant can be used to improve tissue viability when the animal is found dead in decomposition. We have altered the sentence to improve clarity.
Lines 120-123: This sentence should be further expanded. For example, why can’t tissue biopsies be maintained cold under field conditions until processing in the lab? Authors state that ‘require a partnership between different teams and organization…’. What do the authors mean by this? Authors are requested to provide additional clarifications in the manuscript.
Answer: Some of the capture sites are in very remote locations, requiring hikes of several kilometers and without access to any power source. In these cases, collecting material is always even more challenging. We have altered the text in the manuscript to provide additional clarification as requested.
Lines 126-139: As stated earlier, these sentences could be moved to the top of this section to highlight the importance of defining priorities for the biobank and for articulating the long-term objectives of a biobank.
Answer: Thank you for your comment, we hope this version of the manuscript is clearer and the flux of ideas more fluid.
Lines 141-143: The authors state ‘Effective management requires appropriate storage, meticulous sample metadata…’. It would be good to first highlight the importance of metadata and then storage conditions. Further, the authors are also requested to expand on ‘storage’ requirements.
Answer: Thanks for the suggestions. We expanded the text regarding storage requirements and followed up with the metadata information.
Lines 202-204: A major challenge to data sharing is the lack of a universal database that is available to all countries and users. Even in advanced countries such as the United States, not all institutions involved in biobanking use the same database. Recently, zoological information systems (ZIMS; species360.org) has initiated the development of data modules to capture metadata in a standardize manner. However, these resources require annual membership fees that most institutions may not be able to support. Therefore, standardization of metadata and development of an open access (free) database is also warranted.
Answer: Thanks for the amazing comment and suggestion. We have added this perspective into the manuscript.
Lines 209-212: Authors are requested to rewrite this sentence to provide improved clarity.
Answer: We modified this sentence and some terms in the following sentences to improve flow of ideas and logical argument.
Lines 224-231: These statements are not supported by appropriate citations. This reviewer is not aware of any reports where semen from golden lion tamarins have been systematically collected, cryopreserved, and banked. Further, there are no reports of successful development and application of ARTs in this species for species recovery. To this reviewer’s knowledge, the current success was achieved primarily via stringent genetic management and natural breeding. Authors are requested to provide additional evidence to support these statements.
We have added the following reference:
Arakaki, P.R., Salgado, P.A.B., Losano, J.D.D.A., Gonçalves, D.R., Valle, R.D.R.D., Pereira, R.J.G. and Nichi, M., 2019. Semen cryopreservation in golden‐headed lion tamarin, Leontopithecus chrysomelas. American Journal of Primatology, 81(12), p.e23071.
Lines 233-240: The authors seem to over interpret some of the preliminary results that have been published in support of ARTs in bobcats. The mere production of domestic cat x bobcat hybrid embryos does not guarantee the consistent production of biologically competent bobcat embryos when frozen bobcat sperm is used to fertilized bobcat oocytes. Challenges remain with respect to obtaining viable, mature bobcat oocytes consistently. Authors are requested to acknowledge these remaining challenges before claiming that these results could address current challenges in maintaining sustainable populations.
Answer: We have edit adding the following sentence: While these findings represent an important step toward the use of assisted reproductive technologies (ARTs) in wild felids, the results remain preliminary. Significant challenges persist, particularly in obtaining viable and mature bobcat oocytes consistently and in ensuring the production of biologically competent embryos when using conspecific gametes. Therefore, further research is required before these techniques can be reliably applied to support sustainable population management in this species.
Lines 241-248: Authors are requested to discuss potential limitations of dry biobanking as well here. For example, with dry biobanking, more advanced ARTs including ICSI and embryo transfer also must be developed for target species for production of live offspring. This information (and access to ICSI infrastructure) is still lacking for a large number of species and in several countries.
Answer: Thanks for your comment. We have edited the sentence accordingly. One promising example is dry biobanking, an alternative inspired by natural mechanisms such as anhydrobiosis, which involves the preservation of cells through controlled desiccation. This technique could enable the storage of genetic material at room temperature, with lower costs, reduced dependence on specialized infrastructure, and greater safety during transport. Its potential is particularly valuable for countries with technical or economic constraints, as it democratizes access to genetic conservation technologies and expands their global reach [56]. However, the practical application of dry biobanking still faces important limitations. To produce viable offspring from desiccated material, advanced assisted reproductive technologies (ARTs) such as intracytoplasmic sperm injection (ICSI) and embryo transfer must be optimized for each target species. The infrastructure, technical expertise, and species-specific biological knowledge required for these procedures remain limited in many regions, highlighting the need for further research and capacity building before large-scale implementation.
Lines 259-260: The concept of ‘public-private’ partnerships in development of biobanks is often discussed. But it is imperative to discuss potential risks associated with such partnerships. Private entities are often driven by revenue generation or profits. But the use of biomaterials from endangered species for revenue generation raises significant ethical concerns. Emphasis should also be placed on development of strict material transfer agreements to limit the use of these biological materials. Authors are requested to discuss these concerns and suggest solutions to mitigate these concerns in this section.
Answer: We have edit the sentence to address your concerns: o overcome potential ethical risks when profit motives influence the use of biomaterials from endangered species, implementing transparent governance and strict material transfer agreements is essential to prevent misuse and ensure that such collaborations align with conservation goals.

Reviewer 2 Report
Comments and Suggestions for Authors
Biobanking for animal conservation remains a still largely unexploited component of world strategies for conservation of species. This is despite its long term use in plant conservation, animal agriculture and biomedicine. Plus the recognition of its potential, and active technical research, since the late 1970s.
This review very thoroughly covers the opportunities integration of biobanking in wildlife conservation strategies offers and the large number of impediments; technical, legal/political, policy, cultural and funding/economic.
The only deficiency I note is the lack of reference to the economic and genetic modelling of Howell et al that argues strongly for the cost effectiveness of biobanking as an integral part of captive breeding programs as a means to achieve IUCN longterm genetic conservation goals. Arguably the only feasible means.
Although the focus is South America the review is relevant to wildlife conservation globally and thus is of interest to a wide readership
Author Response
Biobanking for animal conservation remains a still largely unexploited component of world strategies for conservation of species. This is despite its long term use in plant conservation, animal agriculture and biomedicine. Plus the recognition of its potential, and active technical research, since the late 1970s.
This review very thoroughly covers the opportunities integration of biobanking in wildlife conservation strategies offers and the large number of impediments; technical, legal/political, policy, cultural and funding/economic.
The only deficiency I note is the lack of reference to the economic and genetic modelling of Howell et al that argues strongly for the cost effectiveness of biobanking as an integral part of captive breeding programs as a means to achieve IUCN longterm genetic conservation goals. Arguably the only feasible means.
Although the focus is South America the review is relevant to wildlife conservation globally and thus is of interest to a wide readership
Answer: We sincerely thank the reviewer for their thoughtful and encouraging comments on our manuscript. In response to the reviewer’s valuable suggestion, we have now included additional references, specifically the work of Howell et al., to address the important aspect of economic and genetic modeling. This addition strengthens our discussion on the cost-effectiveness of biobanking as an integral component of captive breeding programs and its role in achieving long-term genetic conservation goals aligned with IUCN recommendations.
The added references include:
Howell, L.G., Mawson, P.R., Comizzoli, P., Witt, R.R., Frankham, R., Clulow, S., O'Brien, J.K., Clulow, J., Marinari, P. and Rodger, J.C., 2023. Modeling genetic benefits and financial costs of integrating biobanking into the conservation breeding of managed marsupials. Conservation Biology, 37(2), p.e14010.
Howell, L.G., Frankham, R., Rodger, J.C., Witt, R.R., Clulow, S., Upton, R.M. and Clulow, J., 2021. Integrating biobanking minimises inbreeding and produces significant cost benefits for a threatened frog captive breeding programme. Conservation Letters, 14(2), p.e12776.
Howell, L.G., Johnston, S.D., O’Brien, J.K., Frankham, R., Rodger, J.C., Ryan, S.A., Beranek, C.T., Clulow, J., Hudson, D.S. and Witt, R.R., 2022. Modelling genetic benefits and financial costs of integrating biobanking into the captive management of koalas. Animals, 12(8), p.990.

Reviewer 3 Report
Comments and Suggestions for Authors
This is a timely and relevant review that addresses the crucial role of biobanking in conserving South America's exceptional biodiversity. The manuscript effectively synthesizes the key challenges, opportunities, and future directions for biobanking in the region. It highlights important regional case studies and contextualizes the discussion within the unique socio-economic and political landscape of South America. The topic aligns perfectly with the journal's scope. The manuscript is generally well-structured and written. I recommend acceptance after minor revisions to enhance its clarity, depth, and impact.
- The manuscript requires thorough proofreading for minor grammatical errors, typos, and awkward phrasing.And, the "Simple Summary" and "Abstract" are slightly repetitive. Consider refining the abstract to more succinctly state the review's objectives, main findings, and conclusions.
- The section on challenges is good but could be strengthened. When discussing the lack of standardized protocols, briefly mentioning 1-2 key examples of successful protocol adaptations in South America would illustrate the path forward. The critical issue of funding could be expanded upon with more specific suggestions for sustainable models relevant to the region.
- The policy discussion is a key strength. It could be made even more impactful by briefly comparing the regulatory landscape of one or two other South American countries with Brazil and Argentina to provide a more regional overview.
Author Response
This is a timely and relevant review that addresses the crucial role of biobanking in conserving South America's exceptional biodiversity. The manuscript effectively synthesizes the key challenges, opportunities, and future directions for biobanking in the region. It highlights important regional case studies and contextualizes the discussion within the unique socio-economic and political landscape of South America. The topic aligns perfectly with the journal's scope. The manuscript is generally well-structured and written. I recommend acceptance after minor revisions to enhance its clarity, depth, and impact.
Answer: We sincerely thank the reviewer for their positive and encouraging evaluation of our manuscript. In response to the reviewer’s recommendation, we have addressed your comments and hope this versionon is acceptable.
- The manuscript requires thorough proofreading for minor grammatical errors, typos, and awkward phrasing.And, the "Simple Summary" and "Abstract" are slightly repetitive. Consider refining the abstract to more succinctly state the review's objectives, main findings, and conclusions.
Answer: Thank you for your comment. We have revised this section.
- The section on challenges is good but could be strengthened. When discussing the lack of standardized protocols, briefly mentioning 1-2 key examples of successful protocol adaptations in South America would illustrate the path forward. The critical issue of funding could be expanded upon with more specific suggestions for sustainable models relevant to the region.
Answer: We have altered the text to expand on examples. Regarding the funding, we have expanded this topic in the future section.
- The policy discussion is a key strength. It could be made even more impactful by briefly comparing the regulatory landscape of one or two other South American countries with Brazil and Argentina to provide a more regional overview.
Answer: Thank you for your valuable suggestion. We agree that a regional comparison could enhance the discussion; however, the regulatory landscape for biobanking in South America remains largely uncharted, and comprehensive information for many countries is limited. As such, a detailed comparison between Brazil, Argentina, and other countries is currently challenging, but we have clarified this limitation in the revised manuscript to acknowledge the need for further regional analysis.
